# The Changes in the Level of Knowledge about the Effects of Alcohol Use during Pregnancy among Three Last Generations of Women in Poland

**DOI:** 10.3390/ijerph20032479

**Published:** 2023-01-30

**Authors:** Emilia Piotrkowicz, Ilona Kowalik, Iwona Szymusik

**Affiliations:** 1Students Scientific Association at the 1st Department of Obstetrics and Gynecology, First Faculty of Medicine, Medical University of Warsaw, 02-091 Warsaw, Poland; 2National Institute of Cardiology, Clinical Research Support Center, 04-628 Warsaw, Poland; 3The Center of Postgraduate Medical Education, Department of Obstetrics, Perinatology and Neonatology, 01-813 Warsaw, Poland

**Keywords:** pregnancy, alcohol, health promotion, Fetal Alcohol Spectrum Disorders, prenatal alcohol use

## Abstract

Since 1973, when the term Fetal Alcohol Syndrome (FAS) was introduced, a major impact has been put on raising awareness about the negative effects of prenatal alcohol exposure. This study aimed to compare the level of knowledge about the risks of alcohol consumption during pregnancy among three last generations of women in Poland, in order to assess the effectiveness of prenatal education regarding FAS. An online questionnaire was conducted among women of reproductive age, between February and March 2022. The level of knowledge was determined based on the average number of correct answers to 9 questions. Data were analyzed using Cochran–Armitage, ANOVA Kruskal–Wallis, Cochran–Mentel–Haenszel and Pearson’s chi-squared tests. Out of 471 women participating in the study, 34.8% belonged to Generation Z (15–25 years), 55.6% to Generation Y (26–41 years), and 9.6% to Generation X (42–49 years). The average score of correct answers was the highest for Generation Y women (7.55 points) and the lowest for Generation X (6.96 points). Women from Generation Z scored 7.27 points on average. The ANOVA Kruskal–Wallis test was performed with *p* = 0.07. The level of women’s knowledge about the risks of alcohol consumption during pregnancy suggests that education regarding FAS in Poland is less effective in recent years.

## 1. Introduction

For many centuries, scientists had mistakenly assumed that the placenta provides a sufficient barrier between alcohol and a developing fetus. It was not until 1725 that maternal alcohol consumption became associated with fetal neurological anomalies and retarded growth [1]. In 1957, Jacqueline Rouquette studied one hundred children of alcoholic parents, detecting low birth weight, severe mental retardation and congenital malformations [2]. Eleven years later, in 1968, Paul Lemoine published a study of 127 cases of children born to alcoholic mothers, characterizing the constellation of symptoms in their offspring [3]. Both Rouquette’s and Lemoine’s studies, written in French, remained disregarded. The term fetal alcohol syndrome (FAS) was finally introduced in 1973 by Kenneth L. Jones and David W. Smith, who mistakenly claimed to publish “the first reported association between maternal alcoholism and aberrant morphogenesis in the offspring” [4]. The diagnostic criteria of FAS (facial malformations, growth retardation, and central nervous system abnormalities) have remained unchanged since described in 1973 [5].

The very first public awareness warning on FAS was delivered in 1977, by the U.S. Food and Drug Administration. This first advisory recommended a two drink per day limit for pregnant women, and defined six drinks per day as a clear risk of FAS [6]. In 1988, United States Congress passed legislation requiring alcohol-containing beverages to carry a warning label addressing the risks associated with alcohol use in pregnancy [7]. The term fetal alcohol spectrum disorders (FASD) was introduced in 2000 to describe a spectrum of deficits associated with prenatal alcohol exposure [8].

The term FAS was introduced in Poland in 1990s, when the first international publications on this subject were translated and published [9]. After the early 2000s, numerous public health campaigns, including: Pregnancy without alcohol”, were launched in Poland in order to raise FAS awareness [10]. In 2007, the State Agency for Prevention of Alcohol-Related Problems (PARPA) published the first guidelines for medical professionals on monitoring the alcohol use in pregnant women [11]. Since 2008, the Association of Polish Spirits Industry Employers has been conducting informative and educational activities on FAS prevention [9]. According to the Regulation of the Minister of Health from 6 November 2003, it is mandatory to place information about alcohol’s teratogenic effects in beer commercials in Poland [12]. 

In 2012, the Centers for Disease Control and Prevention (CDC) introduced alcohol screening and brief intervention (SBI) into perinatal care [13]. Since then, SBI has become an internationally recommended tool to prevent harm caused by alcohol exposure during pregnancy [14]. Despite those efforts, it was reported in 2012 that over 2% of doctors (including gynecologists and obstetricians) recommended occasional alcohol consumption in pregnancy [15]. The first set of recommendations regarding FAS prevention was submitted in Poland in 2014 [16]. The group of experts stated that pregnant women should avoid consuming any form and any amount of alcohol throughout the entire gestation. Furthermore, they emphasized the role of healthcare professionals and public awareness campaigns in educating future mothers about the consequences of drinking alcohol during pregnancy. 

Despite continuous efforts made by public health authorities to increase FAS awareness, the number of women consuming alcohol during pregnancy remains alarmingly high. Between 2018 and 2019, about 20%–30% of women reported drinking at some point during gestation in the United States, most typically during the first trimester [17]. Those numbers are even higher in the UK, where, in 2017, 41% of pregnant women declared consuming alcohol [18]. As a result, prevalence of the FAS in the United States and some Western European countries might be as high as 2%–5% of the population [19].

The aim of the study was to compare the level of knowledge about the risks of alcohol consumption during pregnancy among the three last generations of women in Poland. A comparison was performed afterwards to assess the effectiveness of current methods for educating future mothers about fetal alcohol spectrum disorders including FAS.

## 2. Materials and Methods

### 2.1. Study Design

The presented study was conducted as an anonymous online questionnaire with 24 closed and 3 open-ended questions. The open-ended questions were as follows: “What’s your religion?”, “What source of information do you consider to be the most reliable when it comes to the effects of alcohol consumption during pregnancy?” and “Were you encouraged to drink alcohol during pregnancy? If yes, who encouraged you?”. Answers to open-ended questions were later assigned as presented in Table 1. 

The survey was posted on Facebook groups for mothers living in different parts of Poland. Data were collected from 3 February 2022 to 3 March 2022. Only completely fulfilled questionnaires were analyzed.

### 2.2. Data Collection

Women of reproductive age (15–49 years), who were Polish citizens and current residents of Poland, were eligible for the study. Respondents were assigned to proper generations based on their birth year: Generation X (women born from 1965 to 1980), Generation Y (women born from 1981 to 1996), and Generation Z (women born from 1997 to 2012).

The survey included questions on socio-demographic data, knowledge about the risks of alcohol consumption during pregnancy and about FAS. Participants who declared being pregnant before or at the time of survey were asked additional questions about their personal experiences regarding alcohol consumption and its prevention during gestation.

### 2.3. Data Analysis

The level of knowledge about the effects of alcohol consumption during pregnancy was assessed based on the number of correct answers to 9 close-ended questions presented in Table 2 (each question valued as 1 point). In order to compare the level of knowledge between Generations X, Y, and Z, the average number of correct answers for women from each of those groups was calculated. 

Statistical significance of acquired data was evaluated with the use of Cochran–Armitage trend test, Cochran–Mentel–Haenszel test, Pearson’s chi-squared test with Yate’s correction, Fisher’s exact test, and ANOVA Kruskal–Wallis test. *p*-values less than 0.1 were considered statistically significant. Statistical analysis was performed with SAS 9.4 Software (SAS Inc., Cary, NC, USA). 

## 3. Results

### 3.1. Characteristics of the Study Group

Briefly, 471 women, aged 15–49, participated in the study. Further, 164 (34.8%) belonged to Generation Z (15–25 years), 262 (55.6%) to Generation Y (26–41 years), and 45 (9.6%) to Generation X (42–49 years). In addition, 257 (54.6%) respondents had been pregnant before. Detailed information about the study group is presented in Table 3. 

### 3.2. Knowledge about the Effects of Alcohol Consumption during Pregnancy

Table 4 presents the number of participants who gave correct answers to questions regarding the level of knowledge about alcohol intake during gestation. The average number of correct answers was the highest among women from Generation Y (7.55 points) and the lowest among women from Generation X (6.96 points). Women from Generation Z scored 7.27 points on average. The ANOVA Kruskal–Wallis test was performed for acquired data, indicating that women from the youngest generation know less about the risks of drinking alcohol during pregnancy than women from Generation Y (*p* = 0.07).

More than 99% of women from the two youngest generations were aware that alcohol is a teratogen, while up to 4.4% of women from Generation X considered alcohol intake during pregnancy harmless. The difference in the knowledge between women from Generation X and those from both younger generations is statistically significant (*p* = 0.047). Furthermore, women from Generation X had a significantly higher alcohol consumption rate during gestation than women from both Generation Y and Generation Z (*p* = 0.031, Table 4). 

The question regarding FAS diagnosis was the most difficult for the surveyed participants. In each generation, almost every fourth woman believes that FAS is always diagnosed directly after childbirth (*p* = 0.78). Only 42.2% of women from Generation X, 48.9% of women from generation Y, and 47.6% of women from Generation Z were aware that FAS can remain undiagnosed for many years (*p* = 0.75). 

Eighty nine percent of women from Generation X and 97% of women from Generation Y had heard about FAS before (*p* = 0.019). Among the youngest generation, only 84% were aware of this condition (*p* < 0.001 in comparison to X + Y generation). 

The percentage of women convinced that only large amounts of alcohol consumed during pregnancy can cause FAS is consistently high: 24% for Generation X, 19% for Generation Y and 22% for Generation Z (and equal in all three groups: *p* = 0.927).

The awareness that abstinence during pregnancy is the only effective method of preventing FAS was the lowest among women from the youngest generation. 

Statistical analysis showed no significant differences between the level of knowledge about the risks of alcohol consumption during pregnancy and: smoking cigarettes (*p* = 0.180), faith and general frequency of alcohol consumption (*p* = 0.490). The number of correct answers depended on the place of residence: women living in the countryside gave the least correct answers, and women living in cities with more than 100,000 inhabitants gave the most correct answers (*p* = 0.032). 

### 3.3. Personal Experiences of Alcohol Exposure Prophylaxis during Pregnancy

The answers to questions concerning personal experiences during pregnancy are presented in Table 5. According to the results of the study, pregnant women from the oldest generation drank alcohol more often than women from Generation Y and Z (24.4% vs. 10.6%; *p* = 0.034). Moreover, with each subsequent generation, the percentage of women encouraged to drink alcohol during gestation by a family member increased. This fact was declared by 2% of women from Generation X, 16% of women from Generation Y, and 20% of women from Generation Z (*p* for trend: 0.028). Seventy six percent of women from Generation X, 81% of women from Generation Y, and 67% of women from Generation Z heard an opinion that drinking a glass of wine occasionally in pregnancy is not harmful to the mother and fetus. According to those results, even though women from the youngest generation encounter this false belief less often than women from both previous generations (80.1%, *p* = 0.002), they are more prone to alcohol consumption during gestation.

### 3.4. The Role of Healthcare Professionals in Preventing Alcohol Exposure during Pregnancy

According to the results, with each subsequent generation, more women are being asked about alcohol consumption during pregnancy and informed about its teratogenic effects by healthcare professionals (4.9% vs. 9.9% vs. 28%; *p* = 0.01). Even though more than 12.2% of Generation X women were encouraged by a healthcare professional to drink alcohol occasionally when pregnant, no woman from Generation Y or Z declared this fact (*p* < 0.001). The level of trust toward healthcare professionals increases with each subsequent generation of women: 35.6% of women from Generation X, 48.9% of women from Generation Y, and 53.0% of women from Generation Z chose healthcare professionals as the most reliable source of information about the effects of alcohol consumption during pregnancy (trend test: *p* = 0.059; X vs. Y + Z: *p* = 0.08). 

## 4. Discussion

A 2012 study by A. Wojtyła et al. showed that around 15% of women in Poland drank alcohol throughout the entire period of pregnancy, even though nearly all of them were aware that alcohol is a teratogen [15]. According to our results, this number is lower in recent years, with 7% of women declaring alcohol consumption at least once during gestation. Furthermore, as our study shows, this problem applies mostly to women aged 42–49, who declare significantly higher alcohol consumption rates during pregnancy than women aged 15–41. Rates of prenatal alcohol exposure in Poland might increase in the future, since, as proven by this study, the level of knowledge about the risks of alcohol exposure during pregnancy is particularly low among the youngest generation of women. According to the presented research, only 84% of women aged 15–25 had heard about conditions such as FAS before. This might prove that the current methods of spreading information about teratogenic effects of alcohol are not adequately adjusted to the needs and expectations of younger generations and should be amended. Further research into this matter is also recommended. 

According to focus research study published by E. Elek et al., healthcare providers remain one of the most important sources of information regarding alcohol intake during gestation for women [20]. Our study shows that the level of trust towards healthcare providers increases with each subsequent generation, with more than 50% of women from Generation Y and Z considering information provided by healthcare professionals as credible. According to the study by J. Elsinga et al., preconceptional counseling can effectively encourage women to reduce alcohol consumption before and during pregnancy [21]. This proves that increasing and updating healthcare providers’ knowledge is crucial in the process of FAS prevention. Regular access to the latest data on this subject should be granted, and international cooperation with centers where scientific research on FAS develops intensively should be undertaken. 

Partners and family members are another important source of information for pregnant women. According to the nationwide survey carried out by the Laboratory for Social Research (PBS) for the State Agency for Prevention of Alcohol-Related Problems (PARPA), 8% of women were encouraged to drink alcohol during pregnancy by a family member mostly [22]. Our study extends those results, showing that with each subsequent generation the percentage of women encouraged by a family member to drink alcohol during gestation increases. As a result, up to 20% of women aged 15–25 declare consuming alcohol during pregnancy, in comparison to 2% of women aged 42–49. Those results show how important it is not only to educate future mothers about the effects of alcohol consumption during gestation, but also the entire society. 

Despite CDC recommendations on alcohol screening in perinatal care, more than 50% of obstetricians in Poland did not discuss alcohol consumption with their pregnant patients in 2012 [15]. The results of the presented study suggest that with each subsequent generation healthcare improves its role in FAS prevention. Women from the youngest generation are asked about alcohol consumption during pregnancy noticeably more often than women from previous generations. The interest in FAS prevention should be increased, especially among representatives of medical specialties, while no official statements and guidelines on this subject have yet been published. Healthcare professionals should be further instructed to discuss the effects of alcohol consumption during pregnancy with their patients, in order to reduce the risk of FAS.

In recent years, numerous scientific publications, public health campaigns, and social events have focused on the harmful effects of alcohol consumption during gestation. According to the study published in 2022 by Okulicz-Kozaryn K, none of the recent Polish campaigns aimed at reducing the risk of alcohol exposure during pregnancy meet the criteria provided by International Standards on Drug Use Prevention [23]. The results of our study are in accordance with Okulicz-Kozaryn data, confirming that current methods of educating women in Poland about FAS are not effective enough. Not only do women aged 15–25 know less about the risks of drinking alcohol during pregnancy than women aged 26–41, but also the level of knowledge about FAS is significantly lower among women living in the countryside than among the ones from major cities. Further research should be conducted to measure the reach and effectiveness of public health campaigns and to develop more effective ways of educating women (especially young women from villages and small towns) about FAS. 

Preventing the consequences of alcohol intake during breastfeeding is another issue crucial for public health. Even though it is well known that consumed alcohol passes freely into the mother’s breast milk, the reports on alcohol intake in lactating women are still scarce [24,25,26]. Current data show that, in developed countries, from 36% to 83% of breastfeeding women drink alcohol [27]. Considering this fact, providing lactating mothers with accurate information on the effects of alcohol consumption is crucial. Since the scope of this study covered only the gestation period, further studies evaluating the level of women’s knowledge about the effects of infants’ exposure to alcohol in mother’s milk should be conducted. 

To our knowledge, this is the first research in Poland comparing the level of FAS awareness between successive generations of women. However, there are some limitations to the study. The presented data are based on self-reported questionnaires, which are liable to recall social desirability bias. However, we believe that the anonymity of participation may minimize the risk of dishonesty and support the concealing of some shameful information, such as alcohol consumption among pregnant women. Nonetheless, there was indeed no way to verify if socio-demographic data declared by participants were genuine. On the other hand, cross-sectional survey-based studies are nowadays widely used to assess public health issues. Due to the widespread access to internet and social media nowadays, such a study design may provide a better representation of the population than personal interviews, leaving the question of the reliability of the results a concern to some extent. 

## 5. Conclusions

In conclusion, women’s knowledge about the risks of alcohol consumption during pregnancy has been decreasing in recent years, despite published recommendations for healthcare providers. This might suggest that current methods of education about fetal alcohol spectrum disorders are not effective enough. The presented study emphasizes the need for raising awareness regarding the effects of alcohol use in pregnancy, especially among the youngest generation of women.

## Figures and Tables

**Table 1 ijerph-20-02479-t001:** Answers assigned to open-ended questions.

Question	Answers
What’s your faith?	Christianity
	Other
Were you encouraged to drink alcohol during pregnancy? If yes, who encouraged you?	I wasn’t encouraged to drink alcohol during pregnancy
	I was encouraged to drink alcohol during pregnancy by a healthcare professional
	I was encouraged to drink alcohol during pregnancy by a family member
	I was encouraged to drink alcohol during pregnancy by somebody else than a healthcare professional or a family member
What source of information do you consider to be the most reliable when it comes to the effects of alcohol consumption during pregnancy?	Family members
	Healthcare professionals
	Media (television and internet)
	Scientific literature or research papers
	Other

**Table 2 ijerph-20-02479-t002:** Questions evaluating the level of knowledge about the effects of alcohol use in pregnancy.

Question	Correct Answer
Do you consider drinking alcohol in pregnancy harmful for the developing fetus?	Yes
Which amount of alcohol do you consider harmful in pregnancy?	Any amount of alcohol use during pregnancy can be harmful to the developing fetus
Which of those types of alcohol is the most dangerous during pregnancy?	All types of alcohol are equally harmful during pregnancy
What causes Fetal Alcohol Syndrome (FAS)?	Consuming any amount of alcohol during pregnancy can cause FAS
Which of those are the symptoms of FAS?	Facial malformations, growth retardation, and central nervous system abnormalities
What characterizes FAS symptoms?	FAS symptoms are permanent and incurable
When is FAS diagnosed?	FAS can remain undiagnosed for many years
Do you agree with the statement that FAS symptoms can differ from one child to another, making it difficult to establish a final diagnosis?	Yes
Is FAS a preventable condition?	Yes, however the only way to prevent FAS is to avoid drinking any alcoholic beverages during pregnancy

**Table 3 ijerph-20-02479-t003:** Socio-demographic characteristics of the study group.

	Generation				
Socio- Demographic Features	All Women *n* = 471	Generation X *n* = 45	Generation Y *n* = 262	Generation Z *n* = 164	*p*-Value
**Educational background**					
Lower secondary education	4 (0.85%)	0 (0.00%)	0 (0.00%)	4 (2.44%)	X vs. Y vs. Z: <0.001 X vs. Z: <0.001 Y vs. Z: <0.001
Secondary education	124 (26.33%)	8 (17.78%)	31 (11.83%)	85 (51.83%)	
Vocational education	2 (0.42%)	0 (0.00%)	0 (0.00%)	2 (1.22%)	
Higher education	341 (72.40%)	37 (82.22%)	231(88.17%)	73 (44.51%)	
**Place of residence**					
Countryside	86 (18.26%)	8 (17.78%)	42 (16.03%)	36 (21.95%)	X vs. Y vs. Z: 0.470
City < 100,000 inhabitants	106 (22.51%)	13 (28.89%)	56 (21.37%)	37 (22.56%)	
City > 100,000 inhabitants	279 (59.24%)	24 (53.33%)	164 (62.60%)	91 (55.49%)	
**Religion**					
Christianity	334 (70.91%)	34 (75.56%)	193 (73.66%)	108 (65.85%)	X vs. Y vs. Z: 0.080
Other	137 (29.09%)	11 (24.44%)	69 (26.34%)		
**Smoking habits**					
Non-smokers	388 (82.38%)	39 (86.67%)	229 (87.40%)	120 (73.17%)	X vs. Y vs. Z: <0.001 Y vs. Z: <0.001 X vs. Z: 0.061
Smokers	83 (17.62%)	6 (13.33%)	33 (12.60%)	44 (26.83%)	
**Frequency of alcohol intake**					
Never or occasionally	179 (38.00%)	21 (46.67%)	107 (40.84%)	51 (31.10%)	X vs. Y vs. Z: 0.029 X vs. Y: 0.109 Y vs. Z: 0.058
Once a month or for a special occasion	172 (36.52%)	13 (28.89%)	87 (33.21%)	72 (43.90%)	
Once a week or more frequently	120 (25.48%)	11 (24.44%)	68 (25.95%)	41 (25.00%)	
**Number of children**					
None	241 (51.17%)	4 (8.89%)	91 (34.73%)	146 (89.02%)	X vs. Y vs. Z: <0.001
One	116 (24.63%)	13 (28.89%)	90 (34.35%)	13 (7.93%)	
Two	87 (18.47%)	20 (44.44%)	63 (24.05%)	4 (2.44%)	
Three or more	27 (5.73%)	8 (17.78%)	18 (6.87%)	1 (0.61%)	

**Table 4 ijerph-20-02479-t004:** The number of correct answers to questions assessing the knowledge about the effects of alcohol consumption during pregnancy for each generation of women.

Question	Generation X *n* = 45	Generation Y *n* = 262	Generation Z *n* = 164	*p*-Value
1. Do you consider drinking alcohol in pregnancy as harmful for the developing fetus?	43 (95.6%)	261 (99.6%)	163 (99.4%)	Trend test: 0.10 X vs. (Y + Z): 0.047
2. Which amount of alcohol do you consider harmful in pregnancy?	43 (95.6%)	257 (98.1%)	158 (96.3%)	Trend test: 0.74 X vs. (Y + Z): 0.36
3. Which of those types of alcohol is the most dangerous during pregnancy?	42 (93.3%)	248 (94.7%)	152 (92.7%)	Trend test: 0.60 X + Y vs. Z: 0.57
4. What causes Fetal Alcohol Syndrome (FAS)?	31 (68.9%)	208 (79.4%)	114 (69.5%)	Trend test: 0.29 Y vs. (X + Z): 0.017
5. Which of those are the symptoms of FAS?	25 (55.6%)	185 (70.6%)	111 (67.7%)	Trend test: 0.43 X vs. (Y + Z): 0.082 Y vs. X: 0.069
6. What characterizes FAS symptoms?	37 (82.2%)	244 (93.1%)	148 (90.2%)	Trend test: 0.49 X vs. (Y + Z): 0.048 Y vs. X: 0.036
7. When is FAS diagnosed?	19 (42.2%)	128 (48.9%)	78 (47.6%)	Trend test: 0.75 X vs. (Y + Z): 0.53
8. Do you agree with the statement that FAS symptoms can differ from one child to another, making it difficult to establish a final diagnosis?	34 (75.6%)	204 (77.9%)	128 (78.1%)	Trend test: 0.78 X vs. (Y + Z): 0.86
9. Is FAS a preventable condition?	39 (86.7%)	243 (92.8%)	141 (86%)	Trend test: 0.23 X vs. (Y + Z): 0.44
10. Have you heard an opinion that drinking alcohol during pregnancy is harmless?	34 (75.6%)	213 (81.3%)	110 (67.1%)	Trend test: 0.015 X + Y vs. Z: 0.002

Trend comparison for 3 groups (X vs. Y vs. Z): Cochran-Armitage Trend Test, comparison 2 groups: chi2 test with Yate’s correction or Fisher’s exact test.

**Table 5 ijerph-20-02479-t005:** Personal experiences during pregnancy.

Question	Generation X *n* = 41	Generation Y *n* = 191	Generation Z *n* = 25	*p*-Value
Consuming alcohol at any time during pregnancy	10 (24.4%)	19 (10.6%)	4 (16.0%)	Trend test: 0.14 X vs. (Y + Z): 0.031
Encouragement to drink alcohol during pregnancy from people other than healthcare professionals	1 (2.4%)	31 (16.2%)	5 (20.0%)	Trend test: 0.028 X vs. (Y + Z): 0.014

## Data Availability

The datasets analyzed in the current study are available from the corresponding author on reasonable request.

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
