# Peer review of "The Changes in the Level of Knowledge about the Effects of Alcohol Use during Pregnancy among Three Last Generations of Women in Poland"

_ijerph, 2023, doi:10.3390/ijerph20032479_

Round 1

Reviewer 1 Report

The Authors presented the level of knowledge about the level of alcohol use among three different generations.

It is  interesting analysis with a conclusion of a need of better education about FAS.

There are however some issues to be explained or corrected.

The shortage FAS and FASD occur throuhtout the article which should be corrected to FAS or FASD.

The questions  and especially the expected answers are not always clear  (maybe its only the matter of translation) i.e. what causes  FAS? the answer: any amount of alcohol and the answer shoud be rather alcohol consumption so I am not sure how the answer was juged as correct or not correct.

I think that the question that FAS symtoms can differ is rather professional one, so random responder may find difficulties in answering.

Author Response

Responses for Reviewer 1

On behalf of all authors I would like to thank you  for your valuable time, comments and suggestions. I would like to address them below:

  1. We decided to use Fetal Alcohol Syndrome definition and shortage FAS, as it is more popular in the literature as well. We are just mentioning in the introduction that the shortage “FASD” is functioning too. The term Fetal Alcohol Spectrum Disorders (FASD) was introduced in 2000 to describe a spectrum of deficits associated with prenatal alcohol exposure.
  2. The problem of the clarity of answers might indeed be related to translation. As to the question “What causes FAS” – the correct answer provided in Table 1 is “Consuming any amount of alcohol during pregnancy can cause FAS” – the word “consuming” is there. Polish respondents were definitely supposed to mark the answer with the word “consumption” and the sentences in Polish language in the survey were simple and clear. They should not have had any problems when deciding upon the chosen answer.
  3. We agree that the question regarding “FAS symptoms can differ…” in Table 1 is rather a difficult one for a random responder, but it is only one out of 9 questions. We assessed the knowledge basing on the average number of correct answers, as we never expected that the surveyed women would answer correct to all of them. We needed easy and difficult questions in order to see any differences in FAS awareness.

All the changes required by the Reviewers that were made in the main text were marked in yellow. All the fragments moved from the discussion to the introduction were marked in green (according to the suggestions of Reviewer 2). I hope our explanations will be satisfactory. Once again thank you for your time.

Yours sincerely,

Iwona Szymusik – corresponding author

Reviewer 2 Report

The article is very interesting, as it highlights a problem that is 100% preventable, such as FASD. I would therefore like to congratulate the authors.

The general approach of the paper is good, although there are some suggestions and changes that I would like to make to the authors for a better understanding of the manuscript.

In the introduction, as usual in works on this subject, it is forgotten that the first description of this pathology is not from 1968, but from 1967, in her doctoral thesis by a forgotten French pediatrician: Jacqueline Rouquette (Rouquette J. Influence de la toxicomanie alcoolique parental sur le développement physique & psychique des jeunes enfants. Université de Paris, Faculté de Médecine. Paris, 1957. [Thèse]. P. 62).

Line 45 introduces some public campaigns launched in Poland, but does not indicate whether other initiatives such as changes in labeling, etc., have also been launched.

Regarding the survey design. It is not indicated which questions are closed and which are open-ended. Likewise, there is no indication of the procedure followed to recode the open questions, whether there was inter-judge agreement, and the values of the indicators of this process.

Regarding the method of data collection, it is not indicated whether the questionnaire included any method of verifying that the respondents were actually women, or whether they were residents of Poland (as stated in lines 65-66). This lack of control in obtaining the sample may be a serious limitation when it comes to taking the results into account. In the same vein, the paper lacks a necessary and important last section on "Limitations".

The results section is quite complete and well understood.

In lines 151-152 there is a sentence that should be translated into English.

I also find the Discussion section problematic as it is organized. Much of it I consider more as part of the introduction than as an integrated discussion of the results obtained. Only the last paragraph refers to the results of the work. There is no explanation of the possible explanation of the differences obtained, nor are possible interventions proposed to alleviate them. I suggest rewriting this section, moving most of it to the introduction.

Finally, I reiterate the lack of a "Limitations" section.

Author Response

Responses for Reviewer 2

On behalf of all authors I would like to thank you  for your valuable time, comments and suggestions. I would like to address them below:

  1. According to your suggestions we moved some parts of the discussion (more general information and parts that were not related directly to our results) to the introduction – they are marked in green in the main text. We hope it will be satisfactory.
  2. We are very grateful for pointing out that we had missed the first description of FAS by Rouquette – we added that valuable information in the introduction and also updated the list of bibliography for the manuscript (changes marked in yellow – new lines 29-41)
  3. We added information when the term FAS was introduced in our country (“The term FAS was introduced in Poland in 1990s, when the first international publications on this subject were translated and published“). It is followed by additional data regarding other initiatives undertaken in Poland (added text in new lines 49-58) – as suggested.
  4. Regarding the survey design: we added some necessary information at the beginning of Material and methods section (new lines 85-90), where we explained more precisely what the open-ended questions were. Additional table (new Table 1) was introduced to show the way of assignment of the answers to open-ended questions as well. We truly hope it will be satisfactory in the presented version. (Previous table 1 is now table 2, and so on with further tables‘ numbers – all changes marked in yellow)
  5. The method of data collection: we tried to address this problem in the suggested section „Limitations“ – thank you very much for this suggestions. It indeed is a very important paragraph in survey-based studies (the last paragraph of the discussion marked in yellow, new lines 275-286). We can never be 100% sure who fulfilled the questionnaire, but we believe that the explanation we provided for the respondents (our information preceeding the survey), where we had pointed out the importance and the purpose of the project at least decreased the bias. We hope our explanation will be satisfactory for you as well.
  6. In the “Results” section of the manuscript we only changed the numeration of tables, as it was necessary.
  7. We are truly grateful for allowing us to improve our discussion. As mentioned above, some general parts of the text were moved to the introduction. We tried to incorporate more information from our study and from the available literature in order to discuss with the obtained results. We rewrote that part of the manuscript and ended it with the limitations of our work, as suggested. All new parts of the discussion are marked in yellow.

Once again thank you for your time and valuable comments. We also thank you for allowing us to revise and resubmit the manuscript. We hope all the provided changes will be satisfactory.

Yours sincerely,

Iwona Szymusik – corresponding author

Round 2

Reviewer 1 Report

I would like to thank the Authors for all explanations

Reviewer 2 Report

 ..